# Pleiotropic Effects of Statins: New Therapeutic Approaches to Chronic, Recurrent Infection by *Staphylococcus aureus*

**DOI:** 10.3390/pharmaceutics13122047

**Published:** 2021-11-30

**Authors:** Melissa D. Evans, Susan A. McDowell

**Affiliations:** Department of Biology, Ball State University, Muncie, IN 47306, USA; melissa.evans96@outlook.com

**Keywords:** statins, ML141, 3-hydroxy-3-methylglutaryl coenzyme A (HMG-CoA), *Staphylococcus* *aureus*, chronic, relapsing, cystic fibrosis, intracellular

## Abstract

An emergent approach to bacterial infection is the use of host rather than bacterial-directed strategies. This approach has the potential to improve efficacy in especially challenging infection settings, including chronic, recurrent infection due to intracellular pathogens. For nearly two decades, the pleiotropic effects of statin drugs have been examined for therapeutic usefulness beyond the treatment of hypercholesterolemia. Interest originated after retrospective studies reported decreases in the risk of death due to bacteremia or sepsis for those on a statin regimen. Although subsequent clinical trials have yielded mixed results and earlier findings have been questioned for biased study design, in vitro and in vivo studies have provided clear evidence of protective mechanisms that include immunomodulatory effects and the inhibition of host cell invasion. Ultimately, the benefits of statins in an infection setting appear to require attention to the underlying host response and to the timing of the dosage. From this examination of statin efficacy, additional novel host-directed strategies may produce adjunctive therapeutic approaches for the treatment of infection where traditional antimicrobial therapy continues to yield poor outcomes. This review focuses on the opportunistic pathogen, *Staphylococcus aureus*, as a proof of principle in examining the promise and limitations of statins in recalcitrant infection.

## 1. Staphylococcus aureus Infection

### Epidemiology

*Staphylococcus aureus* colonizes approximately 30% of the human population, yet colonized individuals typically remain asymptomatic [1,2]. However, *S. aureus* is also an opportunistic pathogen and is the causative agent in life-threatening infections associated with high morbidity and mortality. Wisplinghoff et al. reported *S. aureus* as the second leading cause of bacteremia in hospitals in the United States, exceeded only by coagulase-negative *Staphylococcus* species [3], and Fowler et al. reported *S. aureus* as the leading cause of infective endocarditis worldwide [4]. High mortality rates are associated with *S. aureus* infections. Noskin et al. reported a 5-fold increase in the risk of in-hospital death for *S. aureus* infection compared to non-*S. aureus* infection [5]. Wisplinghoff et al. reported 25% of *S. aureus* bloodstream infections are associated with mortality [3], and De la Calle et al. reported a 30-day mortality rate of 46.9% for *S. aureus* pneumonia [6]. High morbidity and mortality rates are not merely attributable to antibiotic resistance, such as in the reporting of the 19% all-cause in-hospital mortality rate associated with methicillin-susceptible *S. aureus* (MSSA) bloodstream infections [7]. Thus, this opportunistic pathogen inflicts significant morbidity and mortality through infection by resistant strains and by strains susceptible to first-line antimicrobial treatment.

Invasive *S. aureus* strains are an important cause of chronic, relapsing infection, especially notable in cystic fibrosis [8,9]. *S. aureus* is an initial isolate identified in the colonization of the respiratory tract of cystic fibrosis patients, as indicated by Armstrong et al., where 66.6% of infants less than 6 months old with cystic fibrosis had lower respiratory infections caused by *S. aureus* [10]. Evidence that *S*. *aureus* infection persists includes findings from Schwerdt et al. showing 61% of cystic fibrosis patients chronically infected with *S. aureus* for more than 50% of a 22-year observation period [11]. Persistence by the same strain can continue for extended periods, as evidenced in Branger et al., who found 48% of cystic fibrosis patients persistently infected with a single *S. aureus* strain for 12–28 months [12]. The Cystic Fibrosis Foundation Patient Registry annual report for 2019 detailed an increase in the percentage of patients infected with *S. aureus* per year from 56.2% in 2002 to 70.2% in 2019 [13]. Of these infections, 55.3% were attributed to MSSA, more than doubling methicillin-resistant *S. aureus* (MRSA) infection (24.6%). Thus, in addition to acute pathogenesis, both MSSA and MRSA represent an important cause of severe, chronic infections associated with high mortality.

## 2. Role of Host Cell Invasion in Pathogenesis

Conventionally, *S. aureus* had been characterized as an extracellular pathogen; yet more recent evidence indicates a clear role for host cell invasion in pathogenesis. Multiple factors determine the fate of invasive *S. aureus* and the contribution to pathogenesis. The response appears dependent on factors of both the host cell and of the invasive strain. One potential fate is the host cell killing of invading bacteria via the host endocytic pathway [14]. During endocytic uptake by the host cell, the endosomes that encapsulate invading bacteria later fuse with lysosomes to form lysosomes that possess microbicidal properties, including an acidic pH near 5–5.5 and degradative enzymes, such as proteases and lipases. Lysosomes characteristically possess scavenger molecules, including lactoferrin, involved in iron sequestration and the formation of reactive oxygen species. These properties of the lysosome enable the destruction of invading bacteria while sustaining host cell viability during bacterial invasion.

However, invasive staphylococcal strains have been identified that exhibit the capacity to evade lysosomal destruction. Tranchemontagne et al. found that *S. aureus* survives inside macrophages by preventing the formation of the host lysosome [15]. By evading lysosomal destruction, intracellular *S. aureus* can reside within the host cell, protected from antimicrobial therapy, as indicated in the report by Lehar et al., where intracellular *S. aureus* was found to establish infection in a mouse model despite intravenous treatment with the antibiotic vancomycin [16]. Krut et al. found commonly used antibiotics, such as vancomycin and gentamicin, fail to prevent host cell death caused by intracellular *S. aureus* [17]. Invasion into host cells is thought to provide protection from host immune defenses as well [18]. Tuchscherr et al. found *S. aureus* persisting for weeks within host cells in vivo and in vitro, suggesting the insufficiency of the host immune response in clearing intracellular infection. Thus, microbial factors can drive persistence through alteration of the host response.

Invasive *S. aureus* can persist within the host cell as latent bacterial populations [18,19,20]. During intracellular persistence, the host cell remains viable until latent bacteria induce cell death via apoptotic and necrotic mechanisms [20,21,22]. Haslinger-Löffler et al. found *S. aureus* invasion into human peritoneal mesothelial cells induces cell death initially via apoptosis followed by necrosis [22]. Bayles et al. observed *S. aureus* invasion into bovine mammary epithelial cells induces apoptosis as noted by DNA fragmentation and morphological changes associated with apoptosis [21]. Ultimately, the induction of host cell death by invasive *S. aureus* can lead to the release of viable bacteria into the extracellular space. These newly released bacteria can then begin a chronic cycle of relapsing infection by infecting neighboring host cells [18]. *S. aureus* invasion into circulating immune cells provides protection and transport via the bloodstream to initiate infection at secondary sites [23,24,25]. Thus, intracellular infection by *S. aureus* contributes to pathogenesis through apoptotic and necrotic host cell death, resultant tissue damage, and the unleashing of intracellular bacterial populations to spread infection into previously sterile sites.

## 3. Characteristics of Wildtype vs. Small Colony Variants

Clinically, two distinct phenotypes of *S*. *aureus* have been identified: wildtype and a phenotype that develops from and can revert to the wildtype phenotype: small colony variants (SCVs) [18,19,26,27]. Compared to wildtype *S. aureus*, SCVs of *S. aureus* exhibit a slow growth rate and diminished susceptibility to first-line antibiotics oxacillin or vancomycin [19,28,29]. Infection by wildtype *S. aureus* induces the expression of CCL5 and CXCL10, genes involved in the immune response [19]. In contrast, CCL5 and CXCL10 expression remains unchanged in response to SCV *S. aureus* infection. SCVs of *S. aureus* do, however, exhibit an increase in the expression of bacterial adhesion molecules used to invade host cells [18,30]. Tuchscherr et al. reported the increased expression by SCV *S. aureus* of fibronectin-binding protein A (FnBPA), an adhesin important for staphylococcal attachment to the host cell [18]. Vaudaux et al. found that SCVs of *S. aureus* exhibit the increased expression of bacterial FNBPs and bacterial clumping factor A (CflA) and an associated increase in adhesion to host fibrinogen and host fibronectin, which are important for invasion [30].

Both phenotypes can persist intracellularly. Viable wildtype *S. aureus* has been recovered from epithelial cells two weeks after infection [26], from keratinocytes seven days post-infection [27], and from macrophage six days post-infection. SCVs of *S. aureus* can persist in their new cellular niche for extended periods, and upon release, these variant bacteria, expressing high levels of adhesion molecules, rapidly infect neighboring cells [18]. Taken together, both phenotypes successfully invade host cells, establish intracellular populations, and upon release, are capable of infecting naive host cells in a cycle of chronic, recurrent infection.

## 4. Primary Mechanism of Host Cell Invasion

The microbial surface component recognizing adhesive matrix molecule (MSCRAMM) family of proteins is the largest class of *S. aureus* virulence factors involved in adhesion and invasion [31]. The MSCRAMM family is comprised of *S. aureus* surface proteins that mediate binding to host extracellular matrix (ECM) components, including fibrinogen, collagen, and fibronectin. Fibronectin-binding proteins (FnBPs) are members of the MSCRAMM family that bind, with high affinity, to host fibronectin. The major mechanism of *S. aureus* invasion into host cells utilizes FnBPs [31,32,33]. *S. aureus* mutants that do not express FnBPs lose their invasive ability [32]. Furthermore, the ectopic expression of FnBPs in non-invasive bacterial species, such *Staphylococcus carnosus* and *Lactococcus lactis*, confers invasiveness [32,33].

During invasion, FnBPs on the surface of *S. aureus* bind to host fibronectin, which then binds to the host cell integrin receptor α5β1 (Figure 1) [32]. Integrin clustering leads to the activation of the focal adhesion kinase (FAK), a protein tyrosine kinase involved in signal transduction from integrin-enriched adhesion sites [34,35]. The autophosphorylation of FAK leads to SH2 domain-dependent recruitment and binding of the Src family of protein tyrosine kinases [35]. *S. aureus* engagement of the receptor stimulates the FAK–Src complex phosphorylation of cortactin, an actin binding protein [34]. Cortactin is involved in the organization of actin filaments via recruitment of the actin-related protein (Arp)2/3 complex and dynamins [36]. The Arp2/3 complex is involved in actin nucleation and dynamins are involved in the fission of membrane vesicles during receptor-mediated endocytosis. The activation of cortactin by the FAK–Src complex ultimately leads to actin polymerization and endocytosis of the bacteria–fibronectin complex via the Arp2/3 complex and dynamins [34].

The Arp2/3 complex also can be activated indirectly by members of the RHO family of GTPases, including RAC, RHO, and CDC42 [36,37]. Activated RAC, RHO, and CDC42 bind to and activate the Wiskott–Aldrich syndrome protein (WASp) family. Following activation, members of the WASp family bind to and activate the Arp2/3 complex, ultimately leading to actin polymerization. RAC, RHO, and CDC42 function as central regulators of actin stress fiber dynamics [38]. CDC42 can function ahead of RAC and RHO, ultimately leading to the sequential activation of RAC and RHO, respectively. Inactivation of the RHO family of GTPases using *Clostridium difficile* toxin B prevents actin stress fiber polymerization and endocytosis, highlighting the importance of these small GTPases in the regulation of actin dynamics and endocytic uptake [39].

During *S. aureus* invasion into host cells, CDC42 is initially localized at the host cell membrane [40] (Figure 1) and is activated through GTP binding [41,42]. GTP-bound CDC42 has been shown to couple with the p85α isoform of phosphoinositide 3-kinase (PI3K) via the Rho-GAP (also known as the breakpoint cluster region or BCR) homology domain within p85α [43]. Constitutively, PI3Kp85α is bound to available PI3K110 isoforms that carry a catalytic domain [44]. Thus, through this coupling of membrane-bound, GTP-loaded CDC42 with PI3Kp85α, the PI3Kp110 catalytic domain moves from the cytosol to the host cell membrane [43]. At the membrane, the PI3Kp110 catalytic domain can now access membrane-bound phosphoinositide and initiate the formation of the cell signaling molecule, phosphoinositide 3,4,5-trisphosphate (PIP_3_) [44]. Downstream effects are mediated by PIP_3_, including the binding of PIP_3_ to α-actinin, dislodging α-actinin from actin and enabling the reordering of actin subunits [45].

The expression of kinase-dead PI3Kp110α diminishes host cell invasion by *S. aureus*, indicating a role for PI3K catalytic activity in host cell entry [40]. Supporting this, the reversible PI3K inhibitor, LY294002, acting at the ATP binding site within p110 catalytic domains, decreases invasion in a dose-dependent manner. A role for PI3Kp85α in invasion also emerged with the report that the overexpression of mutated PI3Kp85α deficient in the Rho-GAP domain required for coupling to CDC42 decreases host cell invasion [46]. Moreover, the overexpression of mutated PI3Kp85α deficient in GTPase-activating protein (GAP) activity diminishes integrin uptake, integrin receptor recycling, and invasion by *S. aureus* [46]. Based on the central role of this CDC42/PI3K nexus, this pathway may be a viable target for inhibiting bacterial invasion into host cells.

## 5. Treatment Strategies for Intracellular *S. aureus* Infection

Efficacy in killing intracellular infection is limited for first-line, hydrophilic antibiotics that exhibit limited permeability across the host cell membrane [47]. Rifampin, a lipophilic antibiotic, demonstrates a propensity for intracellular uptake and demonstrates efficacy in killing both intracellular and extracellular bacteria [48,49,50,51]. Mandell et al. characterized rifampin’s lipophilicity [51] and efficacy at eradicating intracellular *S. aureus* at reduced concentrations relative to lipophobic antibiotics [50]. Krut et al. observed rifampin eradicated intracellular *S. aureus* by three days post-infection compared to the persistence of viable intracellular bacteria at six days post-infection in clindamycin-, linezolid-, and azithromycin-treated samples [17]. Thus, rifampin has shown promise as an antimicrobial therapeutic targeting intracellular infection.

## 6. Rifampin’s Mechanism of Action

Rifampin prevents bacterial protein synthesis by binding to and inhibiting bacterial RNA polymerase (RNAP) [52]. Bacterial RNAP is composed of a core enzyme and the σ specificity factor [53]. The core enzyme is comprised of five subunits, including two α subunits, a β subunit, a β’ subunit, and an ω subunit. The core enzyme binds to the σ specificity factor to form the RNAP holoenzyme. The σ specificity factor helps with promoter recognition and the initiation of transcription, but when bound to the core enzyme, a portion of the σ specificity factor blocks the RNA exit channel. After the growing RNA strand reaches 11–15 nucleotides, the σ specificity factor dissociates from the core enzyme, leaving the core enzyme to complete elongation of the RNA transcript. Rifampin inhibits the function of bacterial RNAP by binding to the β subunit of the core enzyme [52]. In the presence of rifampin, the initiation of transcription occurs, a single phosphodiester bond is generated, and a dinucleotide is produced. However, rifampin prevents translocation of the RNA dinucleotide, preventing elongation of the RNA chain. The rifampin-mediated prevention of transcription results in impaired protein synthesis and ultimately bacterial cell death.

Although rifampin can penetrate host cells and kill intracellular bacteria, treatment is associated with rapid resistance development and cross-resistance to other antibiotics [54,55,56,57,58,59]. Evidence includes the report by Curry et al., which reported that 36.8% of *C. difficile* isolates were resistant to rifampin [55]. Tajbakhsh et al. found 58.96% of *Mycobacterium tuberculosis* isolates from the Qaem Hospital were rifampin-resistant [58]. Resistance extends to the treatment of *S. aureus* infections, including evidence reported by Zhou et al., where 31.1% of *S. aureus* isolates, over a twelve-month period, were rifampin-resistant [59], and the finding of Bongiorno et al. that 16.4% of MRSA isolates were resistant to rifampin [60].

Mutations in the *rpoB* gene, which encodes the β subunit of bacterial RNAP, confer resistance to rifampin [54,56]. Single amino acid substitutions, deletions, and insertions in the *rpoB* gene have all been implicated in resistance development [52,54,61]. Mutations in the *rpoB* gene produce an altered β subunit that is still able to function in the core enzyme of RNAP but has a decreased affinity for rifampin [62]. Interestingly, mutations in the *rpoB* gene also confer cross-resistance to vancomycin and daptomycin, two last-line antibiotics for treating *S. aureus* infections [54,57]. *rpoB* mutations are associated with a thickening of the cell wall, although the mechanism of thickening remains to be fully characterized [57,63,64]. Increased cell wall thickness appears to reduce the penetration of vancomycin and daptomycin through the cell wall. The reduced penetration of vancomycin is due to the “peptidoglycan-clogging mechanism”. Vancomycin binds to D-alanyl-D-alanine termini of peptidoglycan precursor lipid II molecules at the plasma membrane [65]. This binding prevents the eventual crosslinking of peptidoglycan subunits, preventing cell wall synthesis. The cell wall of viable bacteria contains crosslinked D-alanyl-D-alanine subunits that serve as secondary targets for vancomycin. Increased cell wall thickness results in an increased abundance of vancomycin bound to D-alanyl-D-alanine in the cell wall, forming a physical barrier to vancomycin penetration [66]. The reduced penetration of vancomycin through the cell wall to the lipid II molecules at the plasma membrane enables cell wall synthesis to continue, despite the presence of vancomycin. The reduced penetration of daptomycin through the cell wall is not well understood because the mechanism of action of daptomycin is not well understood. It is thought that daptomycin forms oligomer micelles with Ca^2+^ and these calcium–daptomycin complexes insert into the plasma membrane of the bacteria [67]. The insertion of the calcium–daptomycin complexes into the plasma membrane causes potassium efflux from the cell, membrane depolarization, and cell death. Cui et al. speculate the thickened cell wall, caused by *rpoB* mutations, may serve as an obstacle for the penetration of the large daptomycin–calcium oligomers to the cell membrane [57]. Cross-resistance development is a serious side effect of rifampin treatment that reduces the efficacy of rifampin, vancomycin, and daptomycin antibiotics.

## 7. Rifampin Limitations

Rifampin treatment is associated with severe adverse side effects [56,68,69]. High-dose or long-term rifampin treatment is associated with the development of rifampin-specific antibodies present in the blood [68,69]. Immunologic side effects caused by rifampin range from minor, flu-like symptoms to severe effects, such as thrombocytopenia and acute hemolytic anemia. In rare cases, rifampin treatment has been associated with the development of hepatotoxicity and nephrotoxicity [56].

Due to the development of rifampin resistance and adverse side effects, rifampin is commonly administered with other antibiotics [70]. Rifampin is administered with members of the quinolone family of antibiotics due to the good oral bioavailability of quinolones. There is little in vitro evidence to suggest rifampin cotreatment with other antibiotics enhances bacterial killing; rather, some studies suggest cotreatment may be antagonistic [71,72]. Kaatz et al. observed rifampin antagonized the killing effect of ciprofloxacin on two *S. aureus* strains in vitro [71]. Additionally, Watanakunakorn et al. found rifampin antagonized the killing effect of nafcillin and oxacillin on twenty *S. aureus* strains in vitro [72]. Contrary to in vitro studies, in vivo studies have shown improved bacterial clearance when using rifampin in combination with other antibiotics [73,74]. Greimel et al. found coadministration of rifampin with either moxifloxacin or flucloxacillin reduced the number of viable *S. aureus* relative to monotherapy with any of the three antibiotics [73]. Dworkin et al. found coadministration of rifampin with ciprofloxacin, pefloxacin, and vancomycin increased bacterial clearance relative to monotherapy with ciprofloxacin, pefloxacin, and vancomycin, but did not enhance clearance relative to rifampin monotherapy [74]. Thus, there are conflicting in vitro and in vivo data on the efficacy of rifampin use in conjunction with additional antibiotics.

## 8. Emerging Approaches in Treatment Strategies—Statins

Given the propensity of bacteria, especially *S. aureus*, to develop antibiotic resistance, an emerging approach is to target the host rather than promote resistance by targeting the bacterium [40,41]. One such host-directed approach is the use of statins, therapeutics commonly prescribed in the treatment of hypercholesteremia. Increasing evidence suggests patients prescribed statins for cholesterol-lowering indications exhibit a decreased risk of contracting bacterial infections and improved survival during infections [75,76,77,78,79,80,81,82]. This is supported by findings by Smit et al. that statin-users are 27% less likely to contract community-acquired *S. aureus* bloodstream infections than non-statin users [76]. Almog et al. reported patients on a statin regimen had a 16.6% lower risk of developing sepsis during acute bacterial infections [77]. Björkhem-Bergman et al. found statin use was associated with 50% decreased odds of death during bacterial infections [75]. The protective effect raises the possibility statins may function as a host-directed therapeutic for treating bacterial infections.

However, in a series of randomized clinical trials, statin’s efficacy in treating sepsis or acute respiratory distress syndromes was not supported [83,84,85], calling into question the influence of bias in the earlier retrospective studies [86,87]. Countering this, Kruger et al. [88] demonstrated improved outcomes in patient populations when statin therapy was initiated prior to infection onset. Thus, the timing of statin therapy appears to influence the clinical outcome. The contradiction may signal protection is mediated not only through cholesterol-lowering capacity of statins, but also through pleiotropic effects of statins.

## 9. Statins’ Mechanism of Action

Statins inhibit 3-hydroxy-3-methylglutaryl coenzyme A (HMG-CoA) reductase, the rate limiting enzyme in cholesterol biosynthesis (Figure 2) [89,90,91]. During cholesterol biosynthesis, acetyl-CoA is converted into HMG-CoA by HMG-CoA synthetase [90]. Next, during the rate-limiting step, HMG-CoA reductase converts HMG-CoA into mevalonate, which, through a series of additional steps, is converted into geranyl pyrophosphate and farnesyl pyrophosphate (Fpp). Cholesterol is synthesized from Fpp through numerous additional conversion steps.

## 10. Pleiotropic Effects of Statins

In addition to cholesterol lowering, statins yield cholesterol-independent effects, also known as pleiotropic effects. Statin inhibition of HMG-CoA reductase diminishes synthesis not only of cholesterol, but also of hydrophobic isoprenoid intermediates Fpp and geranylgeranyl pyrophosphate (GGpp) that form from Fpp. Statin pleiotropic effects in part are due to decreased synthesis of these intermediates. These long hydrophobic molecules are modified and attached covalently through the process of post-translational prenylation of proteins containing the conserved CaaX domain [92]. In this domain, “C” is the prenylated cysteine and “X” is the amino acid that determines which isoprenoid, Fpp or GGpp, will become covalently attached to the protein through post-translational prenylation. Thus, when Fpp and GGpp synthesis is reduced through inhibition of the cholesterol biosynthesis pathway by simvastatin, post-translational protein prenylation is likewise reduced [90]. Numerous pleiotropic effects of statins are due to this diminished availability of isoprenoid intermediates.

## 11. Pleotropic Effects of Statins—Inhibition of Infection

Statins interrupt specific stages of host cell invasion through non-cholesterol-dependent pleiotropic effects. The centrality of isoprenoid depletion in these effects is evidenced by the restoration of invasion during simvastatin treatment if Fpp or GGpp are replenished. Invasion was not restored by replenishing cholesterol [40] with this concentration of simvastatin, a lower concentration than that of early work showing apoptotic responses to simvastatin [93]. Multiple effects are rendered through small GTPases, including CDC42, a CaaX domain-containing host cell protein that relies on post-translational prenylation for membrane localization [38,94] (Figure 3). As prenylation decreases following treatment with simvastatin, CDC42 becomes sequestered in the cytoplasm, no longer anchored at the host cell membrane [40]. This loss of membrane localization appears central to several downstream effects.

Although CDC42 remains sequestered within the cytosol, simvastatin stimulates GTP-loading within the CDC42 activation site [46]. In the active, GTP-bound state, mislocated CDC42 is available for coupling with cytosolic PI3Kp85α. Coupled to GTP-bound CDC42 in its cytosolic location, PI3Kp85α is sequestered away from the host cell membrane [40]. The loss of membrane localization of PI3Kp85α potentially results in loss of membrane anchoring for the PI3K110 catalytic subunits that rely on PI3Kp85α for membrane localization. The loss of PI3K110 catalytic subunit access to membrane-bound phosphoinositide would have a resultant loss of formation of the cell-signaling molecule PIP_3_. Evidence of this disruption is that simvastatin treatment limits actin stress fiber disassembly, the endocytic process dependent on PIP_3_ binding α-actinin.

Simvastatin reduces host cell binding to fibronectin [95]. Moreover, simvastatin decreases uptake of the β1 integrin complex from the cell surface and decreases nascent formation of these complexes by limiting recycling of the β1 component to the host cell membrane [46]. Thus, pleiotropic effects of simvastatin, as a host-directed therapeutic, limits *S. aureus* invasion into host cells through decreased synthesis of isoprenoid intermediates, sequestration of RAC, RHO, and CDC42 in the cytosol, decreased membrane localization of these small-GTPases coupled to PI3Kp85α, reduced actin stress fiber depolymerization, decreased host cell binding to fibronectin, and decreased internalization and recycling of β1-integrin receptor complexes to the host cell surface.

In vivo, simvastatin treatment aids in clearing pulmonary infection by invasive *S. aureus* [96]. Similar to findings by Merx et al. in the host response to the endogenous murine microbiome [97,98], simvastatin decreases lung bacterial burden by exogenously administered *S. aureus* and decreases lethality. Simvastatin blunts pulmonary pathogenesis and the inflammatory response to infection, in addition to lowering markers of the inflammatory response both within lung tissue and systemically. Thus, in vivo evidence supports the potential efficacy of statin use for limiting pulmonary infection.

Although pleiotropic effects of statins include inhibition of infection, the use of statins in the critically ill has been challenged due to altered pharmacokinetics within this patient population that renders statins more toxic [99,100,101,102,103]. The usefulness of statins in the treatment of infection has also been questioned following clinical trials demonstrating their limited efficacy [83,84] and concerns that observational studies may have overestimated their therapeutic benefit [86,87]. However, benefit is in evidence in patient populations undergoing statin therapy prior to the onset of infection [80,88,104]. This finding speaks to a potential underlying mechanism of statin efficacy reliant on pleiotropic effects following a reduction in the levels of isoprenoid intermediates that would require a prior statin regimen for efficacy to be achieved.

## 12. Emerging Approaches in Treatment Strategies—ML141

Alternative small molecule inhibitors have been examined that might limit host cell invasion yet circumvent adverse effects and limitations associated with statins. In characterizing the underlying mechanism of simvastatin, RAC, RHO, and CDC42 had emerged as potential molecular targets central to host cell invasion by *S. aureus* [40]. Earlier work had shown CDC42 is activated during *S. aureus* invasion [41] and CDC42 acting upstream of both RAC and RHO [38]. We therefore examined the role of CDC42 by using site-directed mutagenesis to encode valine in place of cysteine within the canonical CAAX prenylation site of CDC42. This inhibition of prenylation within this single host protein diminished invasion by more than 90%, suggesting a central role for CDC42 in invasion [40]. To examine this possibility, we used ML141, a small molecule inhibitor with specificity for hCDC42 [105].

## 13. ML141′s Mechanism of Action

ML141 differs from simvastatin in its target and mechanism of action. While simvastatin demonstrates specificity for HMG-CoA reductase at the early, rate limiting step of cholesterol/isoprenoid biosynthesis and thereby indirectly decreases the prenylation of CDC42, RAC, and RHO [40,106,107], ML141 demonstrates specificity for CDC42 (Figure 4) [38,105]. Acting as an allosteric inhibitor, ML141 dissociates GTP and GDP from the CDC42 active site through rapid, reversible inhibition. Longer-term treatment elicits cellular responses similar to those observed previously in CDC42^-/-^ mouse embryonic fibroblasts (MEF) [108,109]. These downstream effects may be mediated in part through the impaired coupling of GTP-bound CDC42 with downstream effector proteins such as members of the WASp and PI3K families [36,37,40,43,46].

## 14. ML141 Inhibition of Infection

ML141 decreases host cell invasion by MSSA and MRSA strains [41] and by *Streptococcus pyogenes*, a Gram-positive invasive bacterium that shares the fibronectin/integrin invasion mechanism used by *S. aureus* [95]. Similar to simvastatin, the underlying mechanism of inhibition by ML141 includes disruption of α5β1 adhesion complexes at the host cell membrane and decreasing host cell binding to fibronectin (Figure 4). Also similar to simvastatin, ML141 treatment decreases the reordering of actin necessary for endocytic uptake [41]. Thus, although the target and mechanism of action differ between simvastatin and ML141, host cellular responses are similar, contributing to an overall reduction in the establishment of an intracellular bacterial population.

## 15. Emerging Approaches in Treatment Strategies—Antimicrobial Rifampin in Combination with Host-Directed Therapeutics

Given that simvastatin and ML141 decrease the number of bacteria invading host cells and rifampin can kill intracellular bacteria, it was hypothesized the use of rifampin with simvastatin or with ML141 could decrease the number of intracellular bacteria with greater efficacy than rifampin alone (Figure 5). Therefore, a lower concentration of rifampin would be efficacious and advantageous due to high-dose rifampin treatment’s association with severe adverse side effects and resistance development.

To test this hypothesis in vitro, host cells were cotreated with simvastatin and rifampin or with ML141 and rifampin. Interestingly, the response to simvastatin diverged from the response to ML141 [110]. As anticipated, ML141 cotreatment with rifampin decreased intracellular infection more than rifampin alone, nearly doubling the efficacy of antibiotic monotherapy. Conversely, simvastatin cotreatment yielded no detectable improvement in the clearance of intracellular infection.

It is plausible that differential host cell responses are contributing factors in the divergence. Host cell membrane integrity remained intact in response to ML141 yet decreased in response to simvastatin [110]. Cotreatment of rifampin with simvastatin reversed this decrease. The reversal may be due in part to host-directed effects of rifampin. Previous work found rifampin acts not only as an antimicrobial but also induces host cell expression of multidrug resistance protein (MRP) transporters [111]. Members of the MRP transporter family demonstrate an affinity for certain small molecules, including statins, and induce their host cell efflux [112]. Thus, the reversal in host cell membrane permeability may be due to the host-directed, rifampin-induced, efflux of this statin. Although underlying mechanisms remain to be fully characterized, the finding speaks to the complexity of pleiotropic, host-directed effects induced not only by statins, but by antimicrobials as well.

## 16. Summary and Significance

*S. aureus* is an important cause of severe, life-threatening infections and chronic, recurrent infections associated with high morbidity and mortality. Infections are challenging to treat due to multiple factors, including the propensity of *S. aureus* to develop antibiotic resistance and the ability of invasive strains to reside within host cells, protecting the invading bacteria from host immune defenses and antibiotic therapy. Rifampin, although able to penetrate host cells and kill intracellular bacteria, is associated with the development of rapid resistance and with adverse side effects. Emerging evidence indicates the usefulness of alternative strategies that enhance clearance of intracellular infection through host rather than bacterial-directed mechanisms. Statin pleiotropic effects include host-directed protection during non-systemic infection, especially when an ongoing statin regimen can be achieved. However, the novel host-directed small molecule inhibitor ML141 may provide an alternative strategy that circumvents statin limitations by selectively targeting CDC42, a key host cell-signaling molecule implicated in invasive infection. Cotreatment strategies combining host-directed therapeutics with antimicrobials may aid in reducing intracellular bacterial infection and the likelihood of developing antibiotic resistance. Evidence of unintended host responses to both host-directed and antimicrobial therapeutics indicates the complexity of adjunctive therapeutic development. The ongoing challenge of treatment failures in infectious disease necessitates continued expansion of such alternative strategies.

## Figures and Tables

**Figure 1 pharmaceutics-13-02047-f001:**
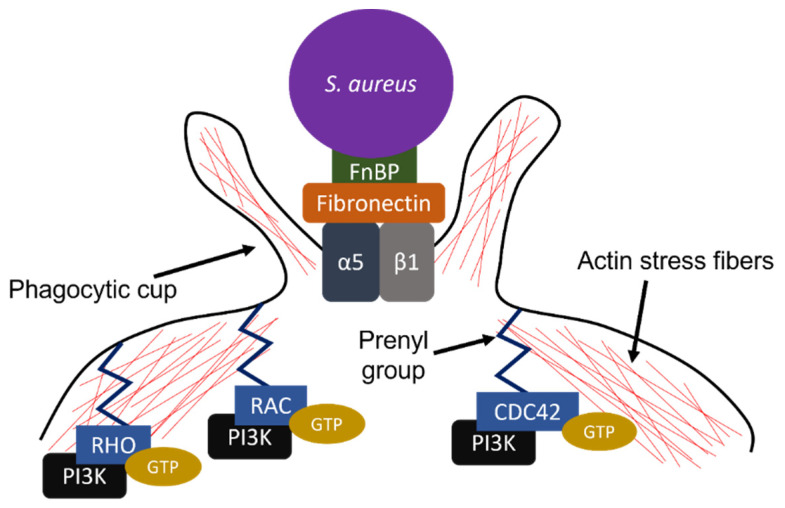
Schematic of *Staphylococcus aureus* invasion into host cells. Fibronectin-binding proteins (FnBPs) on the surface of *S. aureus* bind host extracellular fibronectin and enter the host cell through endocytic uptake of fibronectin bound to cell surface α5β1. During endocytosis, phagocytic cups form through actions of small GTPases RHO, RAC, and CDC42 in concert with phosphoinositide 3-kinase (PI3K).

**Figure 2 pharmaceutics-13-02047-f002:**
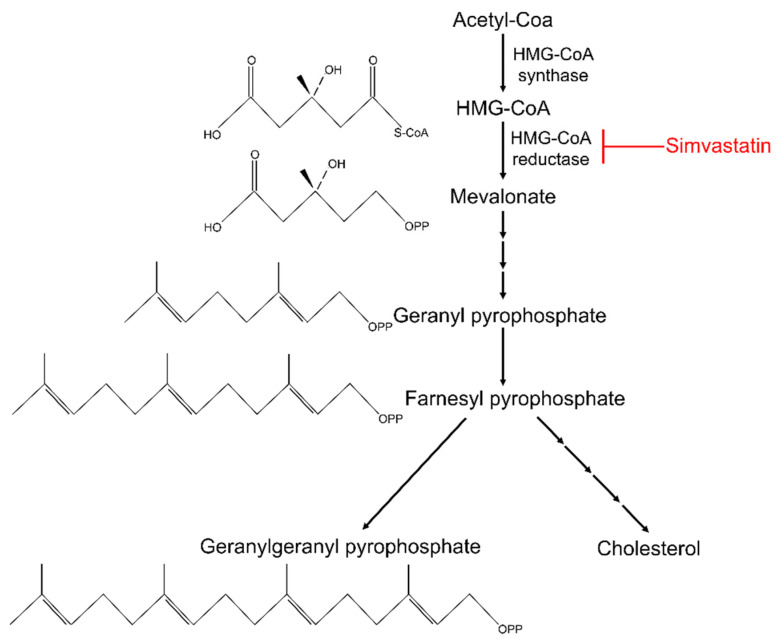
The cholesterol biosynthesis pathway and its inhibition by simvastatin. Acetyl-CoA is converted into 3-hydroxy-3-methylglutaryl coenzyme A (HMG-CoA) by HMG-CoA synthetase. HMG-CoA is then converted into mevalonate by HMG-CoA reductase. Mevalonate is converted into geranyl pyrophosphate and farnesyl pyrophosphate (Fpp). Geranylgeranyl pyrophosphate (GGpp) and cholesterol are both synthesized from Fpp. Simvastatin inhibits HMG-CoA reductase, thus limiting synthesis of these downstream molecules.

**Figure 3 pharmaceutics-13-02047-f003:**
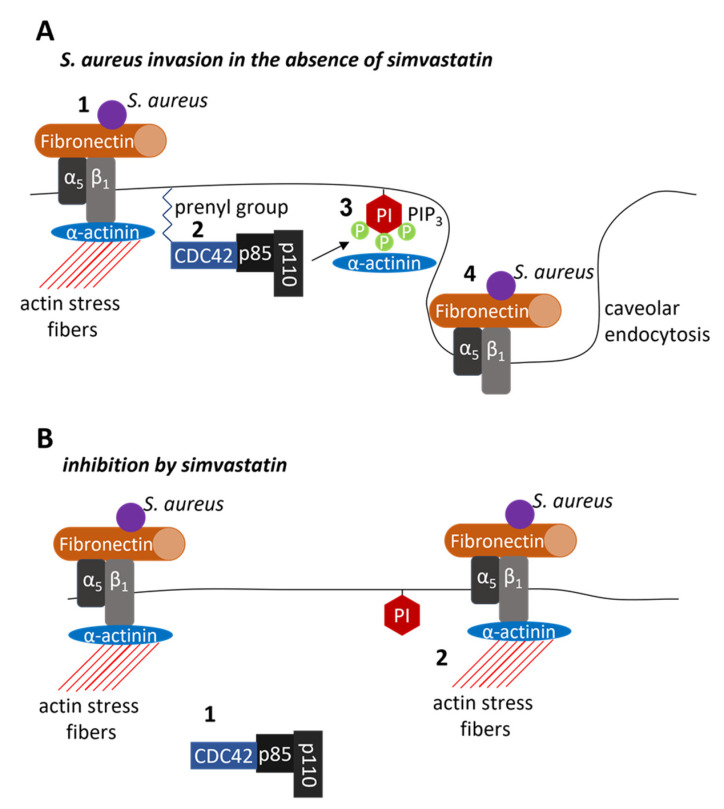
Schematic models of *Staphylococcus aureus* host cell invasion and inhibition by simvastatin. (**A**) **^1^** In the absence of simvastatin, *S. aureus* bound to fibronectin interacts with the integrin α5β1, activating CDC42. **^2^** At the cell membrane, prenylated CDC42 coupled to the p85 subunit of phosphoinositide 3-kinase (PI3K) positions the lipid kinase catalytic domain, p110, in proximity with phosphoinositide (PI). **^3^** The product of PI3K phosphorylation, PI 3,4,5-trisphosphate (PIP_3_), binds to α-actinin, disrupting the interaction with β1 and actin stress fibers. **^4^** Reordering of actin facilitates caveolar endocytosis of the bacterium/fibronectin/integrin complex. (**B**) In response to isoprenoid depletion by simvastatin, **^1^** CDC42 coupled to PI3K accumulates within the cytosol. By sequestering PI3K within the cytosol, CDC42 restricts access to membrane-bound PI. In the absence of PIP_3_, the interaction of α-actinin with β1 and actin stress fibers remain intact **^2^**, removing the pulling forces required for caveolar uptake of the bacterium/fibronectin/integrin complex.

**Figure 4 pharmaceutics-13-02047-f004:**
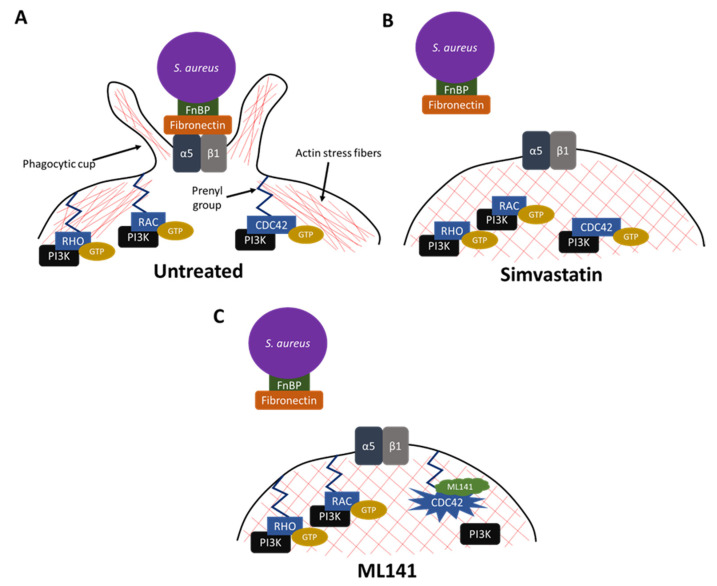
*Inhibition of Staphylococcus aureus invasion during simvastatin and ML141 treatment.* (**A**) Fibronectin-binding proteins (FnBPs) on the surface of *S. aureus* bind host fibronectin. Fibronectin engagement of the host integrin receptor α5β1 stimulates endocytic uptake of the bacterial/fibronectin complex. CDC42, RAC, and RHO are localized at the host cell membrane through long, hydrophobic prenyl anchors. These membrane-bound small GTPases, upon GTP-binding, couple with phosphoinositide 3-kinase (PI3K) and signal disassembly of actin stress fibers and endocytosis. (**B**) During simvastatin treatment, *S. aureus* FnBPs are bound to fibronectin, but there is reduced affinity of α5β1 for fibronectin. CDC42, RAC, and RHO, although bound by GTP, are sequestered with PI3K in the cytosol and actin stress fibers remain intact. (**C**) During ML141 treatment, *S. aureus* FnBPs are bound to fibronectin, but there is reduced affinity of α5β1 for fibronectin. RHO and RAC are bound to GTP. ML141 is bound to CDC42, causing a conformational change and preventing the binding of GTP or GDP to CDC42. Actin stress fibers remain intact.

**Figure 5 pharmaceutics-13-02047-f005:**
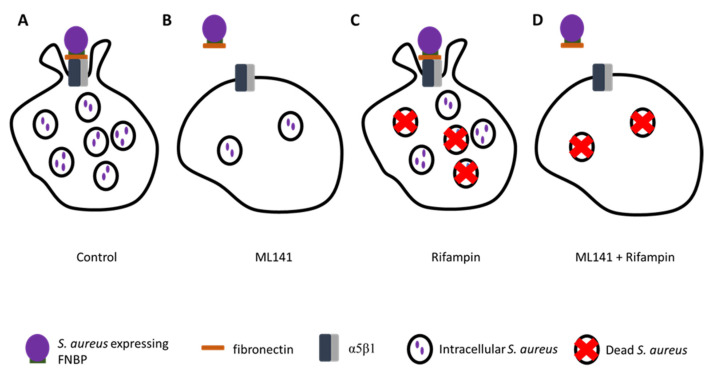
*Schematic of anticipated augmented clearance of intracellular infection in response to cotreatment with ML141 and rifampin.* (**A**) Dimethyl sulfoxide (DMSO)-treated control cells have a large intracellular bacteria population. (**B**) ML141-treated cells have reduced intracellular bacteria due to reduced affinity of fibronectin for α5β1 and decreased depolymerization of actin necessary for endocytic uptake. (**C**) Invasion is occurring in rifampin-treated cells, but there are fewer intracellular bacteria because rifampin is killing some of the bacteria that invade. (**D**) Cotreatment with ML141 and rifampin results in augmented clearance of intracellular bacteria due to reduced numbers of invading bacteria.

## Data Availability

Not applicable.

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
