# Peer review of "Pleiotropic Effects of Statins: New Therapeutic Approaches to Chronic, Recurrent Infection by Staphylococcus aureus"

_pharmaceutics, 2021, doi:10.3390/pharmaceutics13122047_

Round 1

Reviewer 1 Report

  • I believe that the manuscript addresses a current issue such as the effect of statins beyond their cholesterol-lowering effect and in this case addresses the issue of Staphylococcal infection from the perspective of acting on the host instead of on the pathogen in orden to avoid the appearance of resistance.

  • The manuscript is well structured with an initial exposition of the epidemiology of Staphylococcus aureus, then addressing the role of the host in the pathogenesis of this infection, then describing the two different phenotypes of Staphylococcus aureus. It then addresses the treatment strategies for intracellular infection, specifically the role of rifampicin and its limitations, and then goes on to describe the emerging therapies where statin treatment would fit with its mechanism of action and addressing its pleiotropic effect in its role in inhibiting infection by this pathogen. Finally the role of the ML141 molecule in the treatment of this infection es addressed.

  •  

    The manuscript is well expressed, easy to understand and the graphics provide clarity in understanding the mechanisms involved in this infection and the possible treatment of simvastatin in its management.

  • Reviewing the document with the urkund tool, very few similarities are observed with other documents on the same topic.

  •  

    The bibliography is updated with a significant number of citations from the last 5 years and there are no duplications.

  • I believe that the bibliography is cited in a very variable way and therefore all citations should be reviewed and cited with Vancouver standards. In some cases the authors put an author and et al, in others they put several authors ... I think this should be reviewed and put all the bibliography correctly and uniformly.

Reviewer 2 Report

Pleiotropic effects of statins: new therapeutic approaches to chronic, recurrent infection

The authors present an interesting review on pleiotropic effects of statins and new approaches to chronic, recurrent infection.

The review is well written and certainly deserves publication. However, before publications some points should be addressed:

Major issues:

  1. It seems that the paper concerns S. aureus infections, so the title should reflect this fact.
  2. Very important is the issue of S. aureus entry to the host cells: the authors cannot decide whether they discuss endo- or phagocytosis. These are distinct biological processes. It seems that the host-cell entry process is endocytosis (e.g. participation of dynamis) (Agerer et al. cited in the ms).
  3. Considerations on Rifampin action are valuable but do not fit to the intended scope of the paper. Definitely the authors should reconsider the title of their manuscript. In the opinion of this reviewer the subject of the paper is the inhibition of intracellular infection of host cells by S. aureus bacteria.
  4. I am not sure whether inhibition of prenylation of a single protein could be called „pleiotropic effect”.
  5. Line 363: „Statins interrupt multiple stages of host cell invasion through pleiotropic effects….”. This is probably too strong statement!!!
  6. When discussing the differences between mode of action of simvastatin vs ML141 the authors should take into the consideration the effect of cholesterol depletion. This might have a dramatic influence on horizontal heterogeneity of the host cell plasma membrane.

Minor issues,

  1. Line 141: it seems that affinity is high, intermediate or low, but binding is strong or weak.
  2. In Fig. 2, in the formula of HMG and mevalonate the position of H and O should be switched, otherwise H is divalent and O is monovalent.
  3. References look somewhat old. Perhaps the authors should consider limiting citation of less important oler references, so the participation of contemporary literature increases.

Round 2

Reviewer 2 Report

The authors adequately addressed my comments.